# HERE'S MY POINT: ARGUMENTATION MINING WITH POINTER NETWORKS

**Peter Potash, Alexey Romanov & Anna Rumshisky**
Department of Computer Science
University of Massachusetts Lowell
Lowell, MA 01854, USA
{ppotash,aromanov,arum}@cs.uml.edu

## ABSTRACT

One of the major goals in automated argumentation mining is to uncover the argument structure present in argumentative text. In order to determine this structure, one must understand how different individual components of the overall argument are linked. General consensus in this field dictates that the argument components form a hierarchy of persuasion, which manifests itself in a tree structure. This work provides the first neural network-based approach to argumentation mining, focusing on extracting links between argument components, with a secondary focus on classifying types of argument components. In order to solve this problem, we propose to use a modification of a Pointer Network architecture. A Pointer Network is appealing for this task for the following reasons: 1) It takes into account the sequential nature of argument components; 2) By construction, it enforces certain properties of the tree structure present in argument relations; 3) The hidden representations can be applied to auxiliary tasks. In order to extend the contribution of the original Pointer Network model, we construct a joint model that simultaneously attempts to learn the *type* of argument component, as well as continuing to predict links between argument components. The proposed model achieves state-of-the-art results on two separate evaluation corpora. Furthermore, our results show that optimizing for both tasks, as well as adding a fully-connected layer prior to recurrent neural network input, is crucial for high performance.

## 1 INTRODUCTION

Computational approaches to argument mining/understanding have become very popular (Persing & Ng, 2016; Cano-Basave & He, 2016; Wei et al., 2016; Ghosh et al., 2016; Palau & Moens, 2009; Habernal & Gurevych, 2016). One important avenue in this work is to understand the structure in argumentative text (Persing & Ng, 2016; Peldszus & Stede, 2015; Stab & Gurevych, 2016; Nguyen & Litman, 2016). One fundamental assumption when working with argumentative text is the presence of Arguments Components (ACs). The types of ACs are generally characterized as a *claim* or a *premise* (Govier, 2013), with premises acting as support (or possibly attack) units for claims. To model more complex structures of arguments, some annotation schemes also include a *major claim* AC type (Stab & Gurevych, 2016; 2014b).

Generally, the task of processing argument structure encapsulates four distinct subtasks: 1) Given a sequence of tokens that represents an entire argumentative text, determine the token subsequences that constitute non-intersecting ACs; 2) Given an AC, determine the type of AC (*claim*, *premise*, etc.); 3) Given a set/list of ACs, determine which ACs have a link that determine overall argument structure; 4) Given two linked ACs, determine whether the link is of a supporting or attacking relation. In this work, we focus on subtasks 2 and 3.

There are two key assumptions our work makes going forward. First, we assume subtask 1 has been completed, i.e. ACs have already been identified. Second, we follow previous work that assumes a tree structure for the linking of ACs (Palau & Moens, 2009; Cohen, 1987; Peldszus & Stede, 2015; Stab & Gurevych, 2016) Specifically, a given AC can only have a single outgoing link, but can have numerous incoming links. Furthermore, there is a 'head' component that has

First, [cloning will be beneficial for many people who are in need of organ transplants]$_{AC1}$. In addition, [it shortens the healing process]$_{AC2}$. Usually, [it is very rare to find an appropriate organ donor]$_{AC3}$ and [by using cloning in order to raise required organs the waiting time can be shortened tremendously]$_{AC4}$.

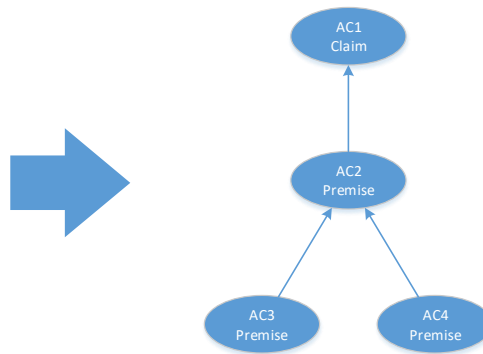

Figure 1: An example of argument structure with four ACs. The left side shows raw text that has been annotated for the presence of ACs. Squiggly and straight underlining means an AC is a *claim* or *premise*, respectively. The ACs in the text have also been annotated for links to other ACs, which is show in the right figure. ACs 3 and 4 are *premises* that link to another *premise*, AC2. Finally, AC2 links to a *claim*, AC1. AC1 therefore acts as the central argumentative component.

no outgoing link (the top of the tree). Figure 1 shows an example that we will use throughout the paper to concretely explain how our approach works. First, the left side of the figure presents the raw text of a paragraph in a persuasive essay (Stab & Gurevych, 2016), with the ACs contained in square brackets. Squiggly verse straight underlining differentiates between claims and premises, respectively. The ACs have been annotated as to how the ACs are linked, and the right side of the figure reflects this structure. The argument structure with four ACs forms a tree, where AC2 has two incoming links, and AC1 acts as the head, with no outgoing links. We also specify the *type* of AC, with the head AC marked as *claim* and the remaining ACs marked as *premise*. Lastly, we note that the order of arguments components can be a strong indicator of how components should related. Linking to the first argument component can provide a competitive baseline heuristic (Peldszus & Stede, 2015; Stab & Gurevych, 2016).

Given the task at hand, we propose a modification of a Pointer Network (PN) (Vinyals et al., 2015b). A PN is a sequence-to-sequence model that outputs a distribution over the encoding indices at each decoding timestep. The PN is a promising model for link extraction in argumentative text because it inherently possesses three important characteristics: 1) it is able to model the sequential nature of ACs; 2) it constrains ACs to have a single outgoing link, thus partly enforcing the tree structure; 3) the hidden representations learned by the model can be used for jointly predicting multiple subtasks. We also note that since a PN is a type of sequence-to-sequence model (Sutskever et al., 2014), it allows the entire sequence to be seen before making prediction. This is important because if the problem were to be approached as standard sequence modeling (Graves & Schmidhuber, 2009; Robinson, 1994), making predictions at each forward timestep, it would only allow links to ACs that have already been seen. This is equivalent to only allowing backward links. We note that we do test a simplified model that only uses hidden states from an encoding network to make predictions, as opposed to the sequence-to-sequence architecture present in the PN (see Section 5).

PNs were originally proposed to allow a variable length decoding sequence (Vinyals et al., 2015b). Alternatively, the PN we implement differs from the original model in that we decode for the same number of timesteps as there are input components. We also propose a joint PN for both extracting links between ACs and predicting the *type* of AC. The model uses the hidden representation of ACs produced during the encoding step (see Section 3.4). Aside from the partial assumption of tree structure in the argumentative text, our models do not make any additional assumptions about the AC types or connectivity, unlike the work of Peldszus (2014). We evaluate our models on the corpora of Stab & Gurevych (2016) and Peldszus (2014), and compare our results with the results of the aforementioned authors.

## 2 RELATED WORK

Recent work in argumentation mining offers data-driven approaches for the task of predicting links between ACs. Stab & Gurevych (2014b) approach the task as a binary classification problem. The

authors train an SVM with various semantic and structural features. Peldszus & Stede (2015) have also used classification models for predicting the presence of links. Various authors have also proposed to jointly model link extraction with other subtasks from the argumentation mining pipeline, using either an Integer Linear Programming (ILP) framework (Persing & Ng, 2016; Stab & Gurevych, 2016) or directly feeding previous subtask predictions into another model. The former joint approaches are evaluated on annotated corpora of persuasive essays (Stab & Gurevych, 2014a; 2016), and the latter on a corpus of microtexts (Peldszus, 2014). The ILP framework is effective in enforcing a tree structure between ACs when predictions are made from otherwise naive base classifiers.

Unrelated to argumentation mining specifically, recurrent neural networks have previously been proposed to model tree/graph structures in a linear manner. Vinyals et al. (2015c) use a sequence-to-sequence model for the task of syntactic parsing. The authors linearize input parse graphs using a depth-first search, allowing it to be consumed as a sequence, achieving state-of-the-art results on several syntactic parsing datasets. Bowman et al. (2015) experiment on an artificial entailment dataset that is specifically engineered to capture recursive logic (Bowman et al., 2014). The text is annotated with brackets, in an original attempt to provide easy input into a recursive neural network. However, standard recurrent neural networks can take in complete sentence sequences, brackets included, and perform competitively with a recursive neural network.

## 3 POINTER NETWORK FOR LINK EXTRACTION

In this section we will describe how we use a PN for the problem of extracting links between ACs. We begin by giving a general description of the PN model.

### 3.1 POINTER NETWORK

A PN is a sequence-to-sequence model (Sutskever et al., 2014) with attention (Bahdanau et al., 2014) that was proposed to handle decoding sequences over the encoding inputs, and can be extended to arbitrary sets (Vinyals et al., 2015a). The original motivation for a pointer network was to allow networks to learn solutions to algorithmic problems, such as the traveling salesperson and convex hull, where the solution is a sequence over candidate points. The PN model is trained on input/output sequence pairs $(E, D)$, where $E$ is the source and $D$ is the target (our choice of $E,D$ is meant to represent the encoding, decoding steps of the sequence-to-sequence model). Given model parameters $\Theta$, we apply the chain rule to determine the probability of a single training example:

$$p(D|E; \Theta) = \prod_{i=1}^{m(E)} p(D_i|D_1, ..., D_{i-1}, E; \Theta) \tag{1}$$

where the function $m$ signifies that the number of decoding timesteps is a function of each individual training example. We will discuss shortly why we need to modify the original definition of $m$ for our application. By taking the log-likelihood of Equation 1, we arrive at the optimization objective:

$$\Theta^* = \arg\max_{\Theta} \sum_{E,D} \log p(D|E; \Theta) \tag{2}$$

which is the sum over all training example pairs.

The PN uses Long Short-Term Memory (LSTM) (Hochreiter & Schmidhuber, 1997) for sequential modeling, which produces a hidden layer $h$ at each encoding/decoding timestep. In practice, the PN has two separate LSTMs, one for encoding and one for decoding. Thus, we refer to encoding hidden layers as $e$, and decoding hidden layers as $d$.

The PN uses a form of content-based attention (Bahdanau et al., 2014) to allow the model to produce a distribution over input elements. This can also be thought of as a distribution over input indices, wherein a decoding step 'points' to the input. Formally, given encoding hidden states $(e_1, ..., e_n)$, The model calculates $p(D_i|D_1, ..., D_{i-1}, E)$ as follows:

$$u^i_j = v^T \tanh(W_1 e_j + W_2 d_i) \tag{3}$$

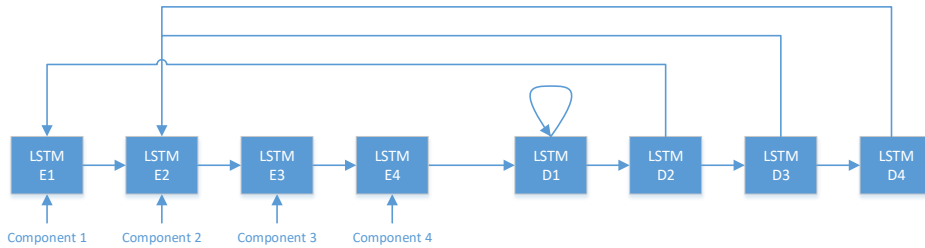

Figure 2: Applying a Pointer Network to the example paragraph in Figure 1 with LSTMs unrolled over time.

$$p(D_i|D_1, ..., D_{j-1}, E) = softmax(u^i) \tag{4}$$

where matrices $W_1$, $W_2$ and vector $v$ are parameters of the model (along with the LSTM parameters used for encoding and decoding). In Equation 3, prior to taking the dot product with $v$, the resulting transformation can be thought of as creating a joint, hidden representation of inputs $i$ and $j$. Vector $u^i$ in equation 4 is of length $n$, and index $j$ corresponds to input element $j$. Therefore, by taking the softmax of $u^i$, we are able to create a distribution over the input.

## 3.2 LINK EXTRACTION AS SEQUENCE MODELING

A given piece of text has a set of ACs, which occur in a specific order in the text, $(C_1, ..., C_n)$. Therefore, at encoding timestep $i$, the model is fed a representation of $C_i$. Since the representation is large and sparse (see Section 3.3 for details on how we represent ACs), we add a fully-connected layer before the LSTM input. Given a representation $R_i$ for AC $C_i$ the LSTM input $A_i$ becomes:

$$A_i = \sigma(W_{rep}R_i + b_{rep}) \tag{5}$$

where $W_{rep}$, $b_{rep}$ in turn become model parameters, and $\sigma$ is the sigmoid function[1]. (similarly, the decoding network applies a fully-connected layer with sigmoid activation to its inputs, see Figure 3). At encoding step $i$, the encoding LSTM produces hidden layer $e_i$, which can be thought of as a hidden representation of AC $C_i$.

In order to make the PN applicable to the problem of link extraction, we explicitly set the number of decoding timesteps to be equal to the number of input components. Using notation from Equation 1, the decoding sequence length for an encoding sequence $E$ is simply $m(E) = |\{C_1, ..., C_n\}|$, which is trivially equal to $n$. By constructing the decoding sequence in this manner, we can associate decoding timestep $i$ with AC $C_i$.

From Equation 4, decoding timestep $D_i$ will output a distribution over input indices. The result of this distribution will indicate to which AC component $C_i$ links. Recall there is a possibility that an AC has no outgoing link, such as if it's the root of the tree. In this case, we state that if AC $C_i$ does not have an outgoing link, decoding step $D_i$ will output index $i$. Conversely, if $D_i$ outputs index $j$, such that $j$ is not equal to $i$, this implies that $C_i$ has an outgoing link to $C_j$. For the argument structure in Figure 1, the corresponding decoding sequence is $(1, 1, 2, 2)$. The topology of this decoding sequence is illustrated in Figure 2. Note how $C_1$ points to itself since it has no outgoing link.

Finally, we note that we modify the PN structure to have a Bidirectional LSTM as the encoder. Thus, $e_i$ is the concatenation of forward and backward hidden states $\overrightarrow{e}_i$ and $\overleftarrow{e}_{n-i+1}$, produced by two separate LSTMs. The decoder remains a standard forward LSTM.

## 3.3 REPRESENTING ARGUMENT COMPONENTS

At each timestep of the decoder, the network takes in the representation of an AC. Each AC is itself a sequence of tokens, similar to the recently proposed Question-Answering dataset (Weston et al., 2015). We follow the work of Stab & Gurevych (2016) and focus on three different types of features

---

[1]We also experimented with relu and elu activations, but found sigmoid to yeild the best performance.

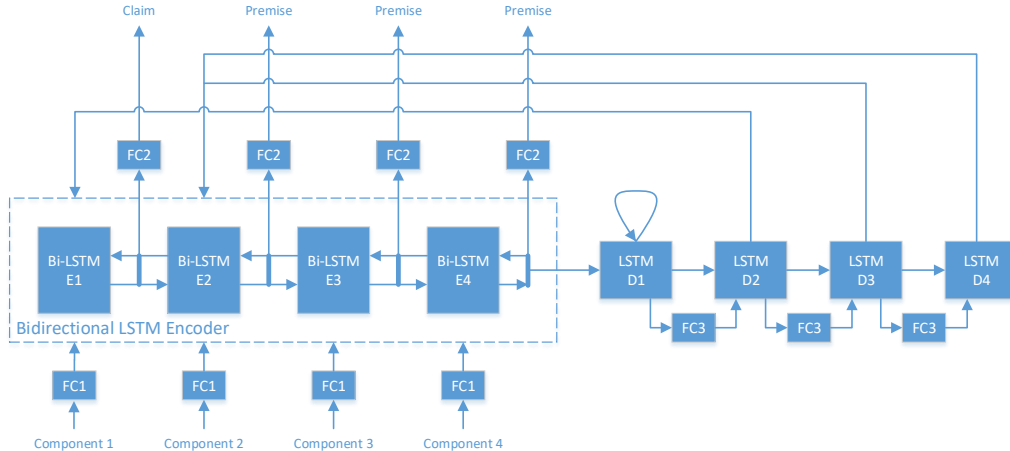

Figure 3: Architecture of the joint model applied to the example in Figure 1.

to represent our ACs: 1) Bag-of-Words of the AC; 2) Embedding representation based on GloVe embeddings (Pennington et al., 2014); 3) Structural features: Whether or not the AC is the first AC in a paragraph, and Whether the AC is in an opening, body, or closing paragraph. See Section 6 for an ablation study of the proposed features.

### 3.4 JOINT NEURAL MODEL

Up to this point, we focused on the task of extracting links between ACs. However, recent work has shown that joint models that simultaneously try to complete multiple aspects of the subtask pipeline outperform models that focus on a single subtask (Persing & Ng, 2016; Stab & Gurevych, 2014b; Peldszus & Stede, 2015). Therefore, we will modify the architecture we proposed in Section 3 so that it would allow us to perform AC classification (Kwon et al., 2007; Rooney et al., 2012) together with link prediction. Knowledge of an individual subtask's predictions can aid in other subtasks. For example, *claims* do not have an outgoing link, so knowing the type of AC can aid in the link prediction task. This can be seen as a way of regularizing the hidden representations from the encoding component (Che et al., 2015).

Predicting AC type is a straightforward classification task: given AC $C_i$, we need to predict whether it is a *claim* or *premise*. Some annotation schemes also include the class *major claim* (Stab & Gurevych, 2014a), which means this can be a multi-class classification task. For encoding timestep $i$, the model creates hidden representation $e_i$. This can be thought of as a representation of AC $C_i$. Therefore, our joint model will simply pass this representation through a fully connected layer as follows:

$$z_i = W_{cls}e_i + b_{cls} \tag{6}$$

where $W_{cls}$, $b_{cls}$ become elements of the model parameters, $\Theta$. The dimensionality of $W_{cls}$, $b_{cls}$ is determined by the number of classes. Lastly, we use softmax to form a distribution over the possible classes.

Consequently, the probability of predicting component type at timestep $i$ is defined as:

$$p(C_i) = p(E_i|\overrightarrow{E}_i, \overleftarrow{E}_i; \Theta) \tag{7}$$

$$p(E_i|\overrightarrow{E}_i, \overleftarrow{E}_i; \Theta) = softmax(z_i) \tag{8}$$

Finally, combining this new prediction task with Equation 2, we arrive at the new training objective:

$$\Theta^* = \arg\max_{\Theta} \alpha \sum_{E,D} \log p(D|E; \Theta) + (1 - \alpha) \sum_{E} \log p(E|\Theta) \tag{9}$$

which simply sums the costs of the individual prediction tasks, and the second summation is the cost for the new task of predicting argument component type. $\alpha \in [0, 1]$ is a hyperparameter that

specifies how we weight the two prediction tasks in our cost function. The architecture of the joint model, applied to our ongoing example, is illustrated in Figure 3.

## 4 Experimental Design

As we have previously mentioned, our work assumes that ACs have already been identified. That is, the token sequence that comprises a given AC is already known. The order of ACs corresponds directly to the order in which the ACs appear in the text. Since ACs are non-overlapping, there is no ambiguity in this ordering. We test the effectiveness of our proposed model on a dataset of persuasive essays (Stab & Gurevych, 2016), as well as a dataset of microtexts (Peldszus, 2014). The feature space for the persuasive essay corpus has roughly 3,000 dimensions, and the microtext corpus feature space has between 2,500 and 3,000 dimensions, depending on the data split (see below).

The persuasive essay corpus contains a total of 402 essays, with a frozen set of 80 essays held out for testing. There are three AC types in this corpus: *major claim*, *claim*, and *premise*. We follow the creators of the corpus and only evaluate ACs within a given paragraph. That is, each training/test example is a sequence of ACs from a paragraph. This results in a 1,405/144 training/test split. The microtext corpus contains 112 short texts. Unlike, the persuasive essay corpus, each text in this corpus is itself a complete example. Since the dataset is small, the authors have created 10 sets of 5-fold cross-validation, reporting the the average across all splits for final model evaluation. This corpus contains only two types of ACs (*claim* and *premise*) The annotation of argument structure of the microtext corpus varies from the persuasive essay corpus; ACs can be linked to other *links*, as opposed to ACs. Therefore, if AC $C_i$ is annotated to be linked to link $l$, we create a link to the source AC of $l$. On average, this corpus has 5.14 ACs per text. Lastly, we note that predicting the presence of links is directional (ordered): predicting a link between the pair $C_i, C_j (i \neq j)$ is different than $C_j, C_i$.

We implement our models in TensorFlow (Abadi et al., 2015). Our model has the following parameters: hidden input dimension size 512, hidden layer size 256 for the bidirectional LSTMs, hidden layer size 512 for the LSTM decoder, $\alpha$ equal to 0.5, and dropout (Srivastava et al., 2014) of 0.9. We believe the need for such high dropout is due to the small amounts of training data (Zarrella & Marsh, 2016), particularly in the Microtext corpus. All models are trained with Adam optimizer (Kingma & Ba, 2014) with a batch size of 16. For a given training set, we randomly select 10% to become the validation set. Training occurs for 4,000 epochs. Once training is completed, we select the model with the highest validation accuracy (on the link prediction task) and evaluate it on the held-out test set. At test time, we take a greedy approach and select the index of the probability distribution (whether link or type prediction) with the highest value.

## 5 Results

The results of our experiments are presented in Tables 1 and 2. For each corpus, we present f1 scores for the AC type classification experiment, with a macro-averaged score of the individual class f1 scores. We also present the f1 scores for predicting the presence/absence of links between ACs, as well as the associated macro-average between these two values.

We implement and compare four types of neural models: 1) The previously described PN-based model depicted in Figure 3 (called PN in the tables); 2) The same as 1), but without the fully-connected input layers; 3) The same as 1), but the model only predicts the link task, and is therefore not optimized for type prediction; 4) A non-sequence-to-sequence model that uses the hidden layers produced by the BLSTM encoder with the same type of attention as the PN (called BLSTM in the table). That is, $d_i$ in Equation 3 is replaced by $e_i$.

In both corpora we compare against the following previously proposed models: Base Classifier (Stab & Gurevych, 2016) is feature-rich, task-specific (AC type or link extraction) SVM classifier. Neither of these classifiers enforce structural or global constraints. Conversely, the ILP Joint Model (Stab & Gurevych, 2016) provides constrains by sharing prediction information between the base classifier. For example, the model attempts to enforce a tree structure among ACs within a given paragraph, as well as using incoming link predictions to better predict the type class *claim*. For the

Table 1: Results on persuasive essay corpus.

| Model | Type prediction | | | | Link prediction | | |
|---|---|---|---|---|---|---|---|
| | Macro f1 | MC f1 | Cl f1 | Pr f1 | Macro f1 | Link f1 | No Link f1 |
| Base Classifier | .794 | .891 | .611 | .879 | .717 | .508 | .917 |
| ILP Joint Model | .826 | .891 | .682 | .903 | .751 | .585 | .918 |
| BLSTM | .810 | .830 | .688 | .912 | .754 | .589 | .919 |
| PN No FC Input | .791 | .826 | .642 | .906 | .708 | .514 | .901 |
| PN No Type | - | - | - | - | .709 | .511 | .906 |
| PN | **.849** | **.894** | **.732** | **.921** | **.767** | **.608** | **.925** |

Table 2: Results on microtext corpus.

| Model | Type prediction | | | Link prediction | | |
|---|---|---|---|---|---|---|
| | Macro f1 | Cl f1 | Pr f1 | Macro f1 | Link f1 | No Link f1 |
| Simple | .817 | - | - | .663 | .478 | .848 |
| Best EG | **.869** | - | - | .693 | .502 | .884 |
| MP+p | .831 | - | - | .720 | .546 | .894 |
| Base Classifier | .830 | .712 | .937 | .650 | .446 | .841 |
| ILP Joint Model | .857 | .770 | .943 | .683 | .486 | .881 |
| PN | .813 | .692 | .934 | **.740** | **.577** | **.903** |

microtext corpus only, we have the following comparative models: Simple (Peldszus & Stede, 2015) is a feature-rich logistic regression classifier. Best EG (Peldszus & Stede, 2015) creates an Evidence Graph (EG) from the predictions of a set of base classifier. The EG models the potential argument structure, and offers a global optimization objective that the base classifiers attempt to optimize by adjusting their individual weights. Lastly, MP+p (Peldszus & Stede, 2015) combines predictions from base classifiers with a MSTParser, which applies 1-best MIRA structured learning.

## 6 DISCUSSION

First, we point out that the PN model achieves state-of-the-art on 10 of the 13 metrics in Tables 1 and 2, including the highest results in all metrics on the Persuasive Essay corpus, as well as link prediction on the Microtext corpus. The performance on the Microtext corpus is very encouraging for several reasons. First, the fact that the model can perform so well with only a hundred training examples is rather remarkable. Second, although we motivate the use of a PN due to the fact that it partially enforces the tree structure in argumentation, other models explicitly contain further constraints. For example, only premises can have outgoing links, and there can be only one claim in an AC. As for the other neural models, the BLSTM model performs competitively with the ILP Joint Model on the persuasive essay corpus, but trails the performance of the PN model. We believe this is because the PN model is able to create two different representations for each AC, one each in the encoding/decoding state, which benefits performance in the dual tasks, whereas the BLSTM model must encode information relating to type as well as link prediction in a single hidden representation. On one hand, the BLSTM model outperforms the ILP model on link prediction, yet it is not able to match the ILP Joint Model's performance on type prediction, primarily due to the BLSTM's poor performance on predicting the *major claim* class. Another interesting outcome is the importance of the fully-connected layer before the LSTM input. The results show that this extra layer of depth is crucial for good performance on this task. Without it, the PN model is only able to perform competitively with the Base Classifier. The results dictate that even a simple fully-connected layer with sigmoid activation can provide a useful dimensionality reduction for feature representation. Finally, the PN model that only extracts links suffers a large drop in performance, conveying that the joint aspect of the PN model is crucial for high performance in the link prediction task.

Table 3 shows the results of an ablation study for AC feature representation. Regarding link prediction, BOW features are clearly the most important, as their absence results in the highest drop in performance. Conversely, the presence of structural features provides the smallest boost in performance, as the model is still able to record state-of-the-art results compared to the ILP Joint Model. This shows that, one one hand, the PN model is able to capture structural ques through sequence

Table 3: Feature ablation study. * indicates that both BOW and Structural are present, as well as the stated embedding type.

| Model | Type prediction | | | | Link prediction | | |
|---|---|---|---|---|---|---|---|
| | Macro f1 | MC f1 | Cl f1 | Pr f1 | Macro f1 | Link f1 | No Link f1 |
| No structural | .808 | .824 | .694 | .907 | .760 | .598 | .922 |
| No BOW | .796 | .833 | .652 | .902 | .728 | .543 | .912 |
| No Embeddings | .827 | .874 | .695 | .911 | .750 | .581 | .918 |
| Only Avg Emb* | .832 | .873 | .717 | .917 | .751 | .583 | .918 |
| Only Max Emb* | .843 | .874 | **.732** | **.923** | .766 | **.608** | .924 |
| Only Min Emb* | .838 | .878 | .719 | .918 | .763 | .602 | .924 |
| All features | **.849** | **.894** | **.732** | .921 | **.767** | **.608** | **.925** |

Table 4: Results of binning test data by length of AC sequence. * indicates that this bin does not contain any *major claim* labels, and this average only applies to *claim* and *premise* classes. However, we do not disable the model from predicting this class: the model was able to avoid predicting this class on its own.

| Bin | Type prediction | | | | Link prediction | | |
|---|---|---|---|---|---|---|---|
| | Macro f1 | MC f1 | Cl f1 | Pr f1 | Macro f1 | Link f1 | No Link f1 |
| $1 \le len < 4$ | .863 | .902 | .798 | .889 | .918 | .866 | .969 |
| $4 \le len < 8$ | .680 | .444 | .675 | .920 | .749 | .586 | .912 |
| $8 \le len < 12$ | .862* | .000* | .762 | .961 | .742 | .542 | .941 |

modeling and semantics (the ILP Joint Model directly integrates these structural features), however the PN model still does benefit from their explicit presence in the feature representation. When considering type prediction, both BOW and structural features are important, and it is the embedding features that provide the least benefit. The Ablation results also provide an interesting insight into the effectiveness of different 'pooling' strategies for using individual token embeddings to create a multi-word embedding. The popular method of averaging embeddings (which is used by Stab & Gurevych (2016) in their system) is in fact the worst method, although its performance is still competitive with the previous state-of-the-art. Conversely, max pooling produces results that are on par with the PN results from Table 1.

Table 4 shows the results on the Persuasive Essay test set with the examples binned by sequence length. First, it is not a surprise to see that the model performs best when the sequences are the shortest. As the sequence length increases, the accuracy on link prediction drops. This is possibly due to the fact that as the length increases, a given AC has more possibilities as to which other AC it can link to, making the task more difficult. Conversely, there is actually a rise in no link prediction accuracy from the second to third row. This is likely due to the fact that since the model predicts at most one outgoing link, it indirectly predicts no link for the remaining ACs in the sequence. Since the chance probability is low for having a link between a given AC in a long sequence, the no link performance is actually better in longer sequences.

# 7 CONCLUSION

In this paper we have proposed how to use a modified PN (Vinyals et al., 2015b) to extract links between ACs in argumentative text. We evaluate our models on two corpora: a corpus of persuasive essays (Stab & Gurevych, 2016), and a corpus of microtexts (Peldszus, 2014). The PN model records state-of-the-art results on the persuasive essay corpus, as well as achieving state-of-the-art results for link prediction on the microtext corpus, despite only having 90 training examples. The results show that jointly modeling the two prediction tasks is crucial for high performance, as well as the presence of a fully-connected layer prior to the LSTM input. Future work can attempt to learn the AC representations themselves, such as in Kumar et al. (2015). Lastly, future work can integrate subtasks 1 and 4 into the model. The representations produced by Equation 3 could potentially be used to predict the *type* of link connecting ACs, i.e. supporting or attacking; this is the fourth subtask in the pipeline. In addition, a segmenting technique, such as the one proposed by Weston et al. (2014), can accomplish subtask 1.

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
