# Peer review of "Here's My Point: Argumentation Mining with Pointer Networks"

_ICLR 2017 — rejected_

[Official Review · AnonReviewer3 · rating 5 · confidence 4 · 11 Dec 2016]
**Solid work, fit unclear**

This paper proposes a model for the task of argumentation mining (labeling the set of relationships between statements expressed as sentence-sized spans in a short text). The model combines a pointer network component that identifies links between statements and a classifier that predicts the roles of these statements. The resulting model works well: It outperforms strong baselines, even on datasets with fewer than 100 training examples.

I don't see any major technical issues with this paper, and the results are strong. I am concerned, though, that the paper doesn't make a substantial novel contribution to representation learning. It focuses on ways to adapt reasonably mature techniques to a novel NLP problem. I think that one of the ACL conferences would be a better fit for this work.

The choice of a pointer network for this problem seems reasonable, though (as noted by other commenters) the paper does not make any substantial comparison with other possible ways of producing trees. The paper does a solid job at breaking down the results quantitatively, but I would appreciate some examples of model output and some qualitative error analysis.

Detail notes: 

- Figure 2 appears to have an error. You report that the decoder produces a distribution over input indices only, but you show an example of the network pointing to an output index in one case.
- I don't think "Wei12" is a name.

[Official Review · AnonReviewer2 · rating 5 · confidence 4 · 13 Dec 2016]

This paper addresses the problem of argument mining, which consists of finding argument types and predicting the relationships between the arguments. The authors proposed a pointer network structure to recover the argument relations. They also propose modifications on pointer network to perform joint training on both type and link prediction tasks. Overall the model is reasonable, but I am not sure if ICLR is the best venue for this work.

My first concern of the paper is on the novelty of the model. Pointer network has been proposed before. The proposed multi-task learning method is interesting, but the authors only verified it on one task. This makes me feel that maybe the submission is more for a NLP conference rather than ICLR. 

The authors stated that the pointer network is less restrictive compared to some of the existing tree predicting method. However, the datasets seem to only contain single trees or forests, and the stack-based method can be used for forest prediction by adding a virtual root node to each example (as done in the dependency parsing tasks). Therefore, I think the experiments right now cannot reflect the advantages of pointer network models unfortunately. 

My second concern of the paper is on the target task. Given that the authors want to analyze the structures between sentences, is the argumentation mining the best dataset? For example, authors could verify their model by applying it to the other tasks that require tree structures such as dependency parsing. As for NLP applications, I found that the assumption that the boundaries of AC are given is a very strong constraint, and could potentially limit the usefulness of the proposed model. 

Overall, in terms of ML, I also feel that baseline methods the authors compared to are probably strong for the argument mining task, but not necessary strong enough for the general tree/forest prediction tasks (as there are other tree/forest prediction methods). In terms of NLP applications, I think the assumption of having AC boundaries is too restrictive, and maybe ICLR is not the best venture for this submission.

[Official Review · AnonReviewer1 · rating 4 · confidence 3 · 19 Dec 2016]
**An Application of PN Network**

This paper addresses automated argumentation mining using pointer network. Although the task and the discussion is interesting, the contribution and the novelty is marginal because this is a single-task application of PN among many potential tasks.

[Author Response · Peter Potash · 20 Jan 2017]
**Full Review Response**

We would like to thank the reviewers for their comments, and specifically, reviewers #2 and #3, who acknowledge that the proposed model is interesting and works well, outperforming strong baselines.  To summarize, the main contribution of our work is to propose a novel joint model based on pointer network architecture which achieves state-of-the-art results on argumentation mining task with a large gap.  

There are three main concerns raised by the reviewers. The first and the main concern is the novelty of the model.  We believe that in part this concern is due to a misunderstanding that occurred because we mislabeled our proposed joint model as “PN” in the results table (see our discussion below).  We think this led some of the reviewers to believe that the paper merely pointer network model to a specific task. 

The second concern is about the overall contribution of the paper to representation learning.  We argue that it is precisely the joint representation learned by our model in the encoding phase, as well as the fact that our model supports separate source and target representations for a given text span, that allows us to substantially outperform standard recurrent models.  

The third question raised by the reviewers concerns the meaningful comparison to other methods for recovering relations, such as stack-based models for syntactic parsing.  We believe that this comparison is not appropriate in our case.  As a discourse parsing task, argumentation mining requires the flexibility of recovering relations that are quite distinct from syntactic parsing, in that they allow both projective and non-projective structures, multi-root parse fragments, and components with no incoming or no outgoing links. 

We give more specific responses to reviewer comments below.

“Pointer network has been proposed before.” 

As is evident in Table 1, a direct application of a pointer network (PN) does not achieve state-of-the-art on link prediction.  Neither is it suitable for AC type prediction.  In fact, a direct application of PN performs substantially worse on link prediction than the joint model, which achieves state-of-the-art on both tasks in the persuasive essay corpus, as well as on link prediction in the microtext corpus. 

“I am concerned, though, that the paper doesn't make a substantial novel contribution to representation learning.” 

We argue that the better performance of the joint PN model is purely due to the representations of the components learned by the model.  When the representation learned in the encoding phase is shared between the two tasks, the model can factor in the information about the types of argument components that are more likely to be linked.  And indeed, our results shows that the information encoded for type prediction is also useful for link prediction, substantially boosting the performance of the joint model.

Furthermore, we also show that the sequence-to-sequence model we propose does much better than standard recurrent models, e.g. Table 1 shows that a BLSTM model can match previous state-of-the-art, which the joint PN model surpasses substantially.  We argue that the better performance of our model is due to the separate representations learned during encoding/decoding. Effectively, our model allows an argument component to have separate representations in its role as a source or a target of a link.

“The proposed multi-task learning method is interesting, but the authors only verified it on one task.” 

Although we do focus on a single task, we test the model on two different datasets, each with its own characteristics. For example, the Microtext corpus has only 100 training examples, and our model is still able to achieve state-of-the-art on link prediction. Furthermore, compared to the Persuasive Essay corpus, the Microtext corpus is much more standardized; all examples are trees and there is exactly one claim in each example. Therefore, even though we focus on a single task, the datasets we test on have varying characteristics, which highlights the generalizability of the model.

“...the stack-based method can be used for forest prediction…” 

Compared to stack-based models, the PN-based framework is more flexible.  Specifically, this framework easily handles projective and non-projective structures, multi-root parse fragments, and components with no incoming or no outgoing links. For example, a stack-based model, such as the one proposed by Dyer et al. (2016), assumes a single-root, projective tree, which is an assumption commonly violated in discourse parsing tasks such as argumentation mining. 

“...I found that the assumption that the boundaries of AC are given is a very strong constraint, and could potentially limit the usefulness of the proposed model.” 

We would like to point out that previous work on AC boundary detection, specifically, the work by the creators of the persuasive essay corpus (Stab & Gurevich 2016) has already achieved near human-level performance in identifying argument components; this is in stark contrast to the previous models for link and type prediction.

“the experiments right now cannot reflect the advantages of pointer network models.” 

We believe this is incorrect.  Specifically, The ILP Joint model uses 18 different hand-engineered features, often requiring external tools such as POS tagging. Our model outperforms it with minimal feature extraction. The MP+p model assumes a single-tree structure explicitly for links (which is unique to the Microtext corpus). Our model outperforms it for link extraction without any explicit single-tree constraint.

[Final Decision · Program Chairs · 06 Feb 2017]
**ICLR committee final decision**

The paper presents an interesting application of pointer networks to the argumentation mining task, and the reviewers found it generally solid. The reviewers generally agree (and I share their concerns) that the contribution on the machine learning side (i.e. the modification to PNs) is not sufficient to warrant publication at ICLR. Moreover, the task is not as standard and extensively studied in NLP; so strong results on the benchmark should not automatically warrant publication at ICLR (e.g., I would probably treat differently standard benchmarks in syntactic parsing of English or machine translation). If judged as an NLP paper, as pointed out by one of reviewers, the lack of qualitative evaluation / error analysis seems also problematic.